# Quantitative SARS-CoV-2 Spike Antibody Response in COVID-19 Patients Using Three Fully Automated Immunoassays and a Surrogate Virus Neutralization Test

**DOI:** 10.3390/diagnostics11081496

**Published:** 2021-08-19

**Authors:** Yoonjoo Kim, Ji Hyun Lee, Geon Young Ko, Ji Hyeong Ryu, Joo Hee Jang, Hyunjoo Bae, Seung-Hyo Yoo, Ae-Ran Choi, Jin Jung, Jongmin Lee, Eun-Jee Oh

**Affiliations:** 1Department of Laboratory Medicine, Seoul St. Mary’s Hospital, College of Medicine, The Catholic University of Korea, Seoul 06591, Korea; malassezia@catholic.ac.kr (Y.K.); hyesungsee@naver.com (J.H.R.); 99056055@cmcnu.or.kr (S.-H.Y.); bibi@cmcnu.or.kr (A.-R.C.); bluejin1227@gmail.com (J.J.); 2Department of Biomedicine & Health Sciences, Graduate School, The Catholic University of Korea, Seoul 06591, Korea; onion1002@naver.com (J.H.L.); geonyoung0107@gmail.com (G.Y.K.); eertrqjjhyu4@naver.com (J.H.J.); jaydcom8673@gmail.com (H.B.); 3Division of Pulmonary, Allergy and Critical Care Medicine, Department of Internal Medicine, Seoul St. Mary’s Hospital, College of Medicine, The Catholic University of Korea, Seoul 06591, Korea; dibs03@gmail.com

**Keywords:** SARS-CoV-2 antibody, chemiluminescent immunoassay, neutralizing antibody, quantitation, binding antibody units

## Abstract

Quantitative SARS-CoV-2 antibody assays against the spike (S) protein are useful for monitoring immune response after infection or vaccination. We compared the results of three chemiluminescent immunoassays (CLIAs) (Abbott, Roche, Siemens) and a surrogate virus neutralization test (sVNT, GenScript) using 191 sequential samples from 32 COVID-19 patients. All assays detected >90% of samples collected 14 days after symptom onset (Abbott 97.4%, Roche 96.2%, Siemens 92.3%, and GenScript 96.2%), and overall agreement among the four assays was 91.1% to 96.3%. When we assessed time-course antibody levels, the Abbott and Siemens assays showed higher levels in patients with severe disease (*p* < 0.05). Antibody levels from the three CLIAs were correlated (r = 0.763–0.885). However, Passing–Bablok regression analysis showed significant proportional differences between assays and converting results to binding antibody units (BAU)/mL still showed substantial bias. CLIAs had good performance in predicting sVNT positivity (Area Under the Curve (AUC), 0.959–0.987), with Abbott having the highest AUC value (*p* < 0.05). SARS-CoV-2 S protein antibody levels as assessed by the CLIAs were not interchangeable, but showed reliable performance for predicting sVNT results. Further standardization and harmonization of immunoassays might be helpful in monitoring immune status after COVID-19 infection or vaccination.

## 1. Introduction

Coronavirus disease 2019 (COVID-19), caused by severe acute respiratory syndrome coronavirus 2 (SARS-CoV-2), has become a pandemic and presents a major health concern across the globe [1,2]. Accurate antibody measurements support uncertain identification or evaluation in the case of resolved infection and can be useful for contact tracing and epidemiologic studies [3,4,5,6]. To date, many SARS-CoV-2 antibody assays have been developed with different antigen targets and assay formats. Most serologic assays are qualitative and use either nucleocapsid (N) or spike (S) SARS-CoV-2 protein as the target for antibody detection. Several studies have already compared some of these assays and found acceptable concordance [6,7,8,9,10]. Recently, quantitative serologic assays for measuring antibodies against the receptor binding domain (RBD) of the S protein have been developed. Quantitative detection may be useful to assist interpretation of COVID-19 immunity and to evaluate active immunization. However, there are a limited number of studies evaluating quantitative S protein antibody levels after COVID-19 infection.

Infection is mediated by interaction of the SARS-CoV-2 S protein RBD with the angiotensin converting enzyme 2 (ACE2) S1 subunit viral receptor on host cells [11]. Antibodies to S protein and RBD can produce a potent virus neutralizing response by inhibiting virus binding to the host ACE2 receptor. With the widespread use of vaccines and therapeutics, longitudinal detection and quantification of antibody responses associated with neutralization becomes increasingly important [12,13]. The SARS-CoV-2 surrogate virus neutralization test (sVNT) (GenScript, Netherlands) is currently available for detecting neutralizing antibodies targeting the RBD based on antibody-mediated blockage of the interaction between the ACE2 receptor and SARS-CoV-2 RBD [14]. To date, limited data are available correlating quantitative SARS-CoV-2 S protein antibody responses with sVNT results [15].

The aim of this study was to evaluate and compare early SARS-CoV-2 S protein antibody responses of COVID-19 patients using three fully automated quantitative chemiluminescent immunoassays (CLIAs): Architect SARS-CoV-2 IgG II Quant (Abbott, Chicago, IL, USA), Elecsys Anti-SARS-CoV-2 S (Roche, Basel, Switzerland), and Atellica IM SARS-CoV-2 IgG (sCOVG) (Siemens, Munich, Germany). We also assessed time-course antibody responses according to disease severity and its correlation with neutralizing antibody results from sVNT.

## 2. Materials and Methods

### 2.1. Patients and Serum Samples

We collected a total of 191 serial serum samples from 32 COVID-19 patients (16 males, 16 females, median age 63 years (range; 35–83 years). All patients were confirmed COVID-19 positive by RT-PCR between March 2020 and December 2020 at Seoul St. Mary’s Hospital. RT-PCRs for detection of SARS-CoV-2 RNA in nasopharyngeal swab samples were performed using the Allplex 2019-nCoV Real-time PCR (Seegene, Seoul, Korea), PowerChek 2019-nCoV (KogeneBiotech, Seoul, Korea), or Real-Q 2019-nCoV Real-Time Detection (BioSewoom, Seoul, Korea) detection kits according to respective manufacturer instructions. Serum remnants were retrieved from blood samples collected for routine laboratory testing during hospitalization and aliquots were stored at −80 °C before analysis. An average of six blood samples (1–15 samples) were retrieved from all patients.

Clinical data for the day after symptom onset and disease course were collected retrospectively from electronic medical records. Patients were classified according to disease course as mild (*n* = 13, nonpneumonia or mild pneumonia), severe (*n* = 14, dyspnea, respiratory frequency ≥30/min, blood oxygen saturation ≤93%, partial pressure of arterial oxygen to fraction of inspired oxygen ratio <300, and/or lung infiltrates >50% within 24 to 48 h), or critical disease (*n* = 5, respiratory failure, septic shock, and/or multiple organ dysfunction or failure) [1]. This study was approved by the Institutional Review Board at Seoul St. Mary’s Hospital (KC20SISI0879). Written informed consent was waived by the board because the current study was retrospective in nature using medical records and residual serum samples.

### 2.2. SARS-CoV-2 Antibody Assays

SARS-CoV-2 S protein antibody levels were measured using three different fully automated chemiluminescent immunoassays (Abbott, Roche, Siemens) and the sVNT (GenScript) according to manufacturer instructions. Detailed descriptions of each assay are shown in Table 1. Samples were retested after additional dilution steps if the measured levels exceeded the measurement limits. We compared qualitative results according to the cut-off values proposed by manufacturers, and also assessed quantitative antibody responses of patients with a critical, severe, or mild disease course. Binding antibody units per milliliter (BAU/mL), which are traceable to WHO international standards for anti-SARS-CoV-2 immunoglobulin, were calculated using conversion factors (Abbott 0.142: Roche 1.028: Siemens 21.803). The correlation between SARS-CoV-2 S protein antibody levels and neutralizing antibody results (%) was also analyzed.

### 2.3. Statistical Analysis

Categorical data are presented as counts and percentage, and continuous data are presented as the median and 95% confidence interval (95% CI). Concordance between assays was calculated using the Cohen Kappa agreement. Kappa values were categorized as slight (0–0.20), fair (0.21–0.40), moderate (0.41–0.60), substantial (0.61–0.80), or excellent (0.81–1.00) [16]. Spearman rank correlation and Passing–Bablok regression were used for comparison of quantitative levels from different assays. The predictive value of antibody assays for sVNT positivity was assessed by the areas under the curve (AUC) from receiver-operating-characteristics (ROC)-curves. All analyses were performed using MedCalc 20.006 (MedCalc, Ostend, Belgium). A *p*-value less than 0.05 was considered statistically significant.

## 3. Results

### 3.1. SARS-CoV-2 Antibody Assay Positivity Rates

Positivity rates of assays in COVID-19 patient samples were assessed. Samples were subdivided into the following groups according to days from symptom onset: ≤5 days, 6–8 days, 9–11 days, 12–14 days, 15–21 days, and ≥22 days. Positivity rates of three CLIAs and the sVNT assay are shown in Figure 1. For specimens collected 6–8 days after symptom onset, positivity rates were 38.5, 42.3, 23.1, and 42.3% for the Abbott, Roche, Siemens, and GenScript assays, respectively. At 15–21 days after symptom onset, Abbott (94.7%), Roche (92.1%) and GenScript (94.7%), except Siemens (86.8%) detected >90% of samples. Overall, for 78 specimens collected 14 days after symptom onset, all four assays detected >90% of samples (Abbott 97.4%, Roche 96.2%, Siemens 92.3%, and GenScript 96.2%).

### 3.2. Agreement between SARS-CoV-2 Antibody Assay Results

Qualitative concordance between assays was determined. Agreement rate was defined as a percentage of samples detected positive or negative in both of two considered immunoassays. Abbott assays showed excellent concordance with Roche (96.3%, k = 0.899) and with Siemens assays (92.1%, k = 0.809). Roche and Siemens assays showed substantial agreement (89.5%, k = 0.739). When we calculated the agreement rates between each CLIA and sVNT, Abbott, Roche, and Siemens assays showed substantial to excellent agreement rates with sVNT at 92.7% (k = 0.807), 93.2% (k = 0.816), and 91.1% (k = 0.785), respectively.

### 3.3. Quantitative SARS-CoV-2 Antibody Levels Related to Days after Symptom Onset and Disease Severity

Antibody levels measured in each immunoassay were compared according to early time-course and disease severity. The measured antibody levels were different depending on the immunoassay. The three assays had different analytical measuring intervals (Abbott, 21.0–40.000 AU/mL; Roche, 0.4–250 U/mL; Siemens, 0.50–150.00 U/mL). The measured levels of 191 specimens ranged up to 35,883.8 AU/mL for Abbott, 3310.0 U/mL for Roche, and 261.6 U/mL for Siemens assays. The results from Roche and Siemens assays exceeded the upper measurement limits and required an additional dilution step. For specimens collected during 15–21 days after symptom onset, the Abbott assay showed the highest antibody levels with a median 95% CI of 4261.5 AU/mL (2769.9–8847.1 AU/mL range), followed by the Roche assay (median 95% CI 155.0 U/mL, 70.3–227.0 U/mL) and the Siemens assay (median 95% CI 37.6 U/mL, 20.2–78.8 U/mL). Neutralizing antibodies levels based on sVNT were 89.0% (84.0–93.0%) during this period. All four antibody assays tended to increase the concentration of severe patients compared to mild patients. When we compared antibody levels in 19 patients with critical or severe course and 13 patients with mild disease course, only the Abbott and Siemens assay results significantly differed (*p* < 0.05, Figure 2, Appendix A).

Early antibody responses were compared according to disease severity using serial samples from each patient. Antibody concentrations in serial serum samples from 32 patients were plotted against days from symptom onset and compared between assays. Antibody kinetics showed high inter-patient variation in concentrations, peak times of antibody levels, and trends over time (Appendix A). Patients with critical or severe disease courses had increased concentrations in three CLIAs compared to mild disease. GenScript assays detecting the neutralizing antibody showed high levels after COVID-19 infection regardless of disease severity.

### 3.4. Correlations between Quantitative SARS-CoV-2 S Protein Antibody Levels from Three Chemiluminescent Immunoassays

The SARS-CoV-2 S protein antibody levels from the three different CLIAs were compared. The antibody levels (U/mL or AU/mL) correlated well as Spearman correlation coefficients for these assays ranged from 0.763 to 0.885 (Appendix A). Disease severity did not affect the correlation coefficients; however, antibody levels were not interchangeable. Passing-Bablok regression analysis showed significant proportional differences: Abbott = −6.8 + 34.1 × Roche, Abbott = −65.5 + 148.9 × Siemens, and Siemens = 0.416 + 0.211 × Roche. The international standard unit (BAU/mL) was calculated using conversion factors provided by manufacturers related to the WHO international standard for SARS-CoV-2 immunoglobulin. As shown in Figure 3, the calculated BAU/mL levels from the three CLIAs showed significant bias in the Passing–Bablok regression. Significant deviations from linearity were found between the three CLIAs (*p* < 0.05).

### 3.5. Correlation between Binding Antibody Values and Neutralizing Antibody Results

We compared antibody levels from the three CLIAs to neutralizing antibody results of sVNT (Figure 4). Overall, log transformed concentration levels of the three CLIAs showed strong correlation with inhibition results (%) from sVNT (r = 0.917–0.945). When we performed ROC analysis of CLIAs for predicting sVNT positivity, AUC values were 0.987 for Abbott, 0.972 for Roche, and 0.959 for Siemens assays. Among the three CLIAs, Abbott had the highest AUC value compared to Roche (*p* = 0.032) or Siemens (*p* = 0.002).

## 4. Discussion

In the present study, three fully automated SARS-CoV-2 antibody CLIA assays widely available to many medical laboratories were compared focusing on quantitative measurement of SARS-CoV-2 S protein RBD antibodies during the early infection period. Expanding testing capacity with validated quantitative measurement is critical to addressing the ongoing pandemic and demonstrating vaccine success [17,18]. Overall positivity rates for the three binding antibody assays and one sVNT differed depending on the days after symptom onset, but positivity rates were comparable. In the early period (6–8 days after symptom onset), positivity rates were from 23.1% to 42.3%. For samples collected at more than 14 days after symptom onset, positivity rates ranged from 92.3% to 97.4%. The positivity rate of the Siemens assay was lower with 92.3% in samples 14 days after symptom onset compared to Abbott (97.4%) and Roche assays (96.2%). This finding is consistent with a previous comparison showing lower sensitivity of the qualitative Siemens assay [19]. While we report excellent positivity rates, our results differed from manufacturer claimed sensitivities (91.1% to 99.4%). This may be due to the characteristics of the patient population. Overall, the three CLIAs showed concordance between assays (k = 0.739–0.899), possibly due to the same assay platform and the same target antigen.

Next, we compared quantitative antibody levels according to time course. All assays showed peak levels around 2 to 3 weeks after symptom onset. This finding confirms previous reports that the median concentrations of antibodies rapidly increased up to 20 days and antibody kinetics of coronavirus predicted a peak at around 2 to 4 weeks followed by a subsequent decrease of antibody titer [20,21]. However, the measured antibody levels differed depending on the assay kit and disease severity. These findings confirm previous reports showing heterogeneity in antibody responses in qualitative and quantitative serologic assays [8,9,22]. Additionally, differences of quantitative levels could be a result of the different measurement ranges of the three assays. In the present study, the measured levels exceeded the upper measurement limits for both the Roche and Siemens assays. According to recent reports measuring antibody levels after vaccination [23,24], the antibody levels for fully vaccinated individuals are expected to be significantly higher than those for infected patients as observed in our study. Therefore, the upper limit of measurement will be frequently exceeded in most analyses. Clinical laboratories should be aware of the range of assay measurements and consider additional dilution steps as required. When we assessed the sequential antibody responses in 32 patients, different antibody kinetics depending on the assay kit were revealed, even when testing with samples from the same patient. This finding confirms previous reports revealing inter-individual differences in SARS-CoV-2 antibody responses [9,25,26].

All four antibody test kits tended to detect increased antibody levels in patients with severe versus mild disease. This trend is in line with previous reports demonstrating earlier seroconversion and high titer for severe disease, but later seroconversion and low titer for mild disease and asymptomatic patients by assays targeting S or RBD protein [8,26,27]. In the subgroup analysis according to days from symptom onset, the Abbott and Siemens assays showed statistically significant higher levels in patients with severe disease course (*p* < 0.05). Although the Roche and sVNT assays showed a tendency of higher levels in severe disease infection, no statistical significance was found. Further evaluations with a large number of specimens are needed for clarification.

Next, we evaluated the correlation between the three CLIAs for S protein antibody quantitation. The three CLIAs showed statistically significant correlation (r = 0.763–0.885); however, antibody levels are not interchangeable, which leads to the inference that direct comparison of numerical results from different test systems is not possible. International Standards allows accurate calibration of assays to an arbitrary unit, thus reducing inter-laboratory variation and leading to harmonization of immune monitoring assessment [28]. Reporting immune responses against the International Standard is crucial for the evaluation of clinical data. Recently, the first WHO international SARS-CoV-2 antibody standard with a value of 1000 BAU/mL was introduced, and manufacturers suggested the conversion factors of U/mL in BAU/mL. However, in present study, the calculated levels (BAU/mL) using conversion factors still showed significant difference and systemic bias based on Passing–Bablok regression analysis. This may be due to subsequent calculation instead of calibration using standard materials. Among the three CLIAs, the least proportional errors were observed between the Abbott and Siemens IgG levels.

Humoral immune response mediated by antibodies is critical to preventing viral infections. Therefore, the most useful information regarding SARS-CoV-2 antibody binding assays is the correlation between antibody values and protective immunity [29]. The current gold standard is the conventional virus neutralization test, which requires a live pathogen and a biosafety level 3 laboratory, and sVNT has shown a good correlation with neutralizing antibody titer by sVNT [13,30]. In the present study, we found a good qualitative agreement of SARS-CoV-2 antibody results from CLIAs with sVNT, confirming previous results [9,15]. When we correlated log transformed concentrations of the three CLIAs with neutralizing antibodies results (%), strong correlations (r = 0.917–0.945) were found. In ROC curve analysis, all three assays showed good performance (AUC 0.959–0.987) for detecting neutralizing antibodies by sVNT.

This study has several limitations. First, this study includes a relatively small number of samples from hospitalized patients, and serum samples of asymptotic patients were not included. Second, it was impossible to show longitudinal results and waning antibody response. In addition, we did not specifically demonstrate the performance of CLIAs to monitor antibody response after vaccination. Monitoring the response to vaccines with neutralization assays would be more clinically useful to assess antibody response and vaccine efficacy. Despite these limitations, our results compare SARS-CoV-2 antibody concentrations from three widely available immunoassays and correlated the antibody response with disease severity and neutralizing antibody results from sVNT. Further studies are needed to assess the performance of different quantitative assays in longitudinal analysis.

## 5. Conclusions

In conclusion, quantitative SARS-CoV-2 S protein antibody levels were not interchangeable but showed a strong correlation and reliable performance for predicting sVNT results in the early COVID-19 infection period. Quantitative measurements of antibody levels will be useful to monitor the course of the immune response against SARS-CoV-2 and active immunization in detail. Further harmonization of antibody assays is required to standardize the assessment of immune response and the degree of protection. Authors should discuss the results and how they can be interpreted from the perspective of previous studies and of the working hypotheses. The findings and their implications should be discussed in the broadest context possible. Future research directions may also be highlighted.

## Figures and Tables

**Figure 1 diagnostics-11-01496-f001:**
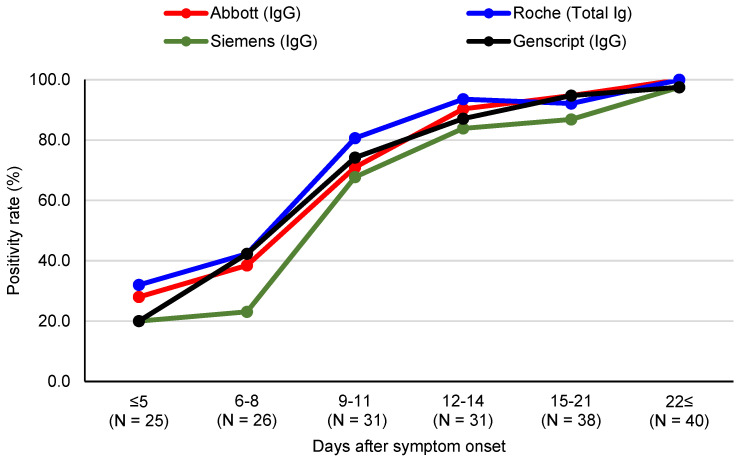
Positivity rate of three SARS-CoV-2 S protein antibody chemiluminescent assays and the surrogate virus neutralization test according to days after symptom onset.

**Figure 2 diagnostics-11-01496-f002:**
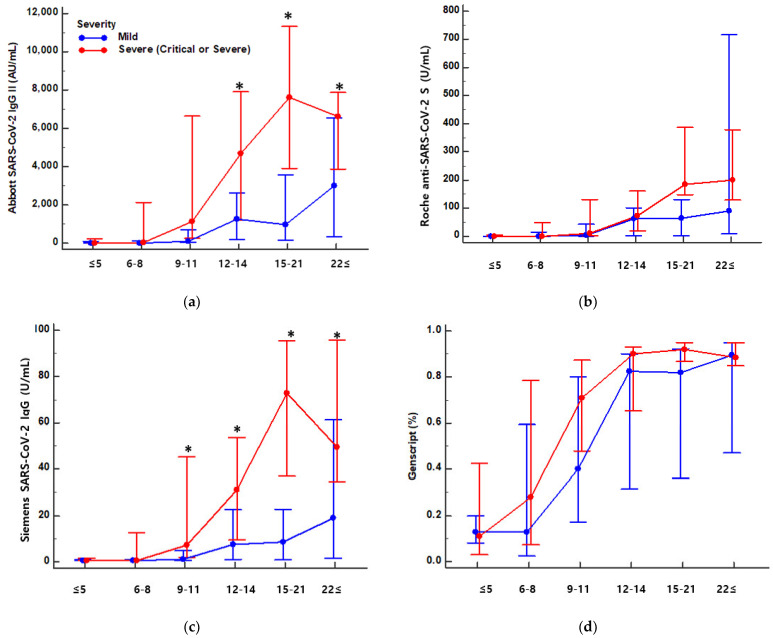
SARS-CoV-2 S protein antibody levels by three chemiluminescent immunoassays and the surrogate virus neutralization test in COVID-19 patients according to days after symptom onset and disease severity. (**a**) Abbott, (**b**) Roche, (**c**) Siemens and (**d**) Genscript. Patients with severe (critical or severe) and mild disease courses are indicated in red and blue, respectively. (* *p* < 0.05).

**Figure 3 diagnostics-11-01496-f003:**
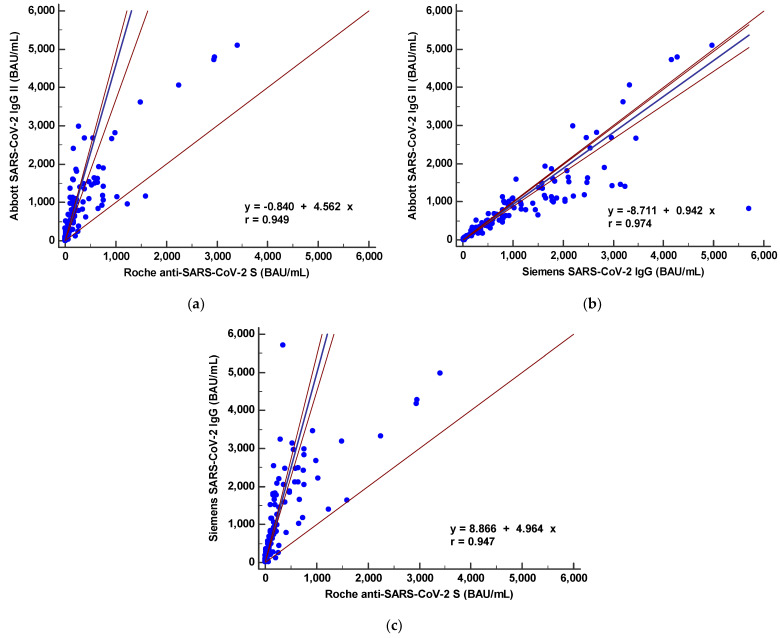
Comparison between binding antibody levels (converted to binding antibody unit (BAU)/mL) from three SARS-CoV-2 S protein antibody immunoassays by Passing–Bablok regression analysis. (**a**) Abbott vs. Roche, (**b**) Abbott vs. Siemens and (**c**) Siemens vs. Roche.

**Figure 4 diagnostics-11-01496-f004:**
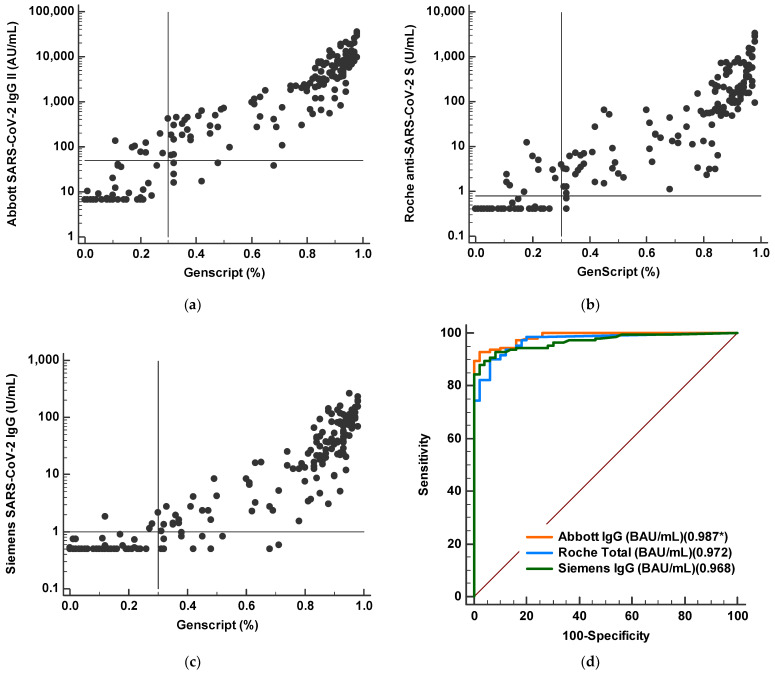
Comparison of quantitative SARS-CoV-2 S protein antibody levels and neutralizing antibody values by surrogate virus neutralization test. (**a**–**c**) Correlations between quantitative concentrations by three CLIAs and percent inhibition by sVNT. (**d**) Comparison of ROC curve analysis to predict sVNT positivity (>30%) for three CLIAs (Abbott, Roche, and Siemens assays). * AUC value.

**Table 1 diagnostics-11-01496-t001:** Characteristics of three SARS-CoV-2 S antibody assays.

	SARS-CoV-2 IgG II Quant	Elecsys Anti-SARS-CoV-2 S	SARS-CoV-2 IgG (sCOVG)
Manufacturer	Abbott Diagnostics	Roche Diagnostics	Siemens
Target antigen	S RBD	S RBD	S1 RBD
Isotype	IgG	Total Ab	IgG
Principle	chemiluminescent microparticle immunoassay (CMIA)	electrochemiluminescence immunoassay (ECLIA)	Chemiluminescence immunoassay (CLIA)
Used analyzer	Architect i2000	Cobas e801	Atellica IM
Calibration	4 parameter logistic curve fit data reduction	2-point calibration	2-point calibration
Specimen	Serum, Dipotassium EDTA Tripotassium EDTA Lithium heparin, Sodium heparin ACD, Sodium citrate	Serum, Li-heparin, EDTA and sodium citrate plasma.	Serum and plasma (lithium heparin)
Required sample volume	75 μL	12 μL	40 µL
Interpretation of results	Positive: ≥50.0 AU/mL	Positive: ≥0.80 U/mL	Reactive: ≥1.00 index (U/mL)
Analytical measuring interval	21.0–40,000 AU/mL	0.40–250 U/mL	0.50–150.00 index
reportable range	6.8–80,000 AU/mL	Not suggested	Not suggested
Limit of blank	5.7	0.3	0.4
Limit of detection	6.8	0.35	0.5
Limit of quantitation	21	0.4	0.5
automated dilution protocol	1:2	1:10	1:5

## Data Availability

The data presented in this study are available on request from the corresponding author.

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
