# Peer review of "Quantitative SARS-CoV-2 Spike Antibody Response in COVID-19 Patients Using Three Fully Automated Immunoassays and a Surrogate Virus Neutralization Test"

_diagnostics, 2021, doi:10.3390/diagnostics11081496_

Round 1

Reviewer 1 Report

The aim of this manuscript was to study and compare three automated quantitative chemiluminescent immunoassays (CLIAs) in the detection of SRAS-CoV-2S protein induced antibody in 32 COVID-19 positive patients. A time course antibody response was measured by using these methods, and these results were compared with surrogate virus neutralization test (sVNT).  The author concluded that these three methods showed a strong correlation with sVNT result in early COVID-19 infection period.

This study is interesting as it was the first study to quantitative measure COVID-19 antibody using these three automated methods.  This study would be useful in the future for researchers and physicians to monitor COVID-19 patient prognosis and management treatment.

Minor comments:  

  1. In Fig. 2, it is not clear of how to define the Mild VS Severe COVID-19 symptoms in these patients. It would be useful for the authors to list the criteria.
  2. Following the comment No.1, the author should compare antibody level using these kits in patient with different symptoms as listed.
  3. In Fig. 4 the Y-axis label is missing in A-C.

Author Response

Reviewer #1: 

In Fig. 2, it is not clear of how to define the Mild VS Severe COVID-19 symptoms in these patients. It would be useful for the authors to list the criteria.

Answer; Thank you for careful review, according to the reviewer’s comment, we added the criteria of severity to the Materials and Methods section as “Patients were classified according to disease course as mild (n = 13, nonpneumonia or mild pneumonia), severe (n = 14, dyspnea, respiratory frequency ³30/min, blood oxygen saturation ≤93%, partial pressure of arterial oxygen to fraction of inspired oxygen ratio <300, and/or lung infiltrates >50% within 24 to 48 hours), or critical disease (n = 5, respiratory failure, septic shock, and/or multiple organ dysfunction or failure)” in lines 79-84.

Following the comment No.1, the author should compare antibody level using these kits in patient with different symptoms as listed.

Answer; In accordance with the reviewer’s suggestion, we added a comparison of antibody levels to Table S1 (Supplementary Materials).

In Fig. 4 the Y-axis label is missing in A-C.

Answer; We have changed Figures.

Reviewer 2 Report

The paper entitled “Quantitative SARS-CoV-2 spike antibody response in COVID-19 patients using three fully automated immunoassays and a surrogate virus neutralization test” evaluates and correlates three fully automated CLIAs for measuring SARS-CoV-2 antibodies and a surrogate virus neutralization test using samples from patients with different levels of disease severity and collected over a period ranging from symptom onset up to 35-40 days. This represents an interesting topic given the importance to both monitor the course of immune response after the infection and evaluate or demonstrate the efficacy of the current vaccines.

The paper is well written and comprehensive, and the main results are clear.

Several minor points should be addressed.

Pag 2 line 71: rephrase sentence.

Number of patients with critical, severe and mild disease course should be defined.

Table 1, please explain the data regarding the correlation to viral neutralization for all three CLIAs and add references accordingly.

Figure 1 Positivity rate at 15-21 days after symptom onset doesn’t seem to reach value >90 for Siemens as stated in the text (pag 4 lines 120-121).

Pag 4 line 133: “Abbott, Roche, and Siemens assays showed substantial to excellent agreement rates with sVNT…”

Pag 4 line 140: “Abbott, 5.8-40.000 AU/mL” should be changed with “Abbott, 21.0-40,000 AU/mL”

Figure 2: add the p value or star (*) in both Abbott and Siemens graphs to indicate the statistical significance.

Figure 2: did “severe” samples include “critical” samples?

Figure 4:  the y-axis is missing label

Pag 8 line 213 the concordance between the assays ranged from 0.739 to 0.899, 0.816 represents the agreement rate between Siemens and sVNT.

Author Response

Reviewer #2:

Pag 2 line 71: rephrase sentence.

Answer; Thank you for careful review. Follwing the reviewer’s comment, we rephrased it as “RT-PCRs for detection of SARS-CoV-2 RNA in nasopharyngeal swab samples were per-formed using the Allplex 2019-nCoV Real-time PCR (Seegene, Seoul, Korea), PowerChek 2019-nCoV (KogeneBiotech, Seoul, Korea), or Real-Q 2019-nCoV Real-Time Detection (Bi-oSewoom, Seoul, Korea) detection kits according to respective manufacturer instructions.” in lines 71-74.

Number of patients with critical, severe and mild disease course should be defined.

Answer; According to the reviewer’s comment, we added the criteria of severity in Materials and Methods section as “Patients were classified according to disease course as mild (n = 13, nonpneumonia or mild pneumonia), severe (n = 14, dyspnea, respiratory frequency ³30/min, blood oxygen saturation ≤93%, partial pressure of arterial oxygen to fraction of inspired oxygen ratio <300, and/or lung infiltrates >50% within 24 to 48 hours), or critical disease (n = 5, respiratory failure, septic shock, and/or multiple organ dysfunction or failure)” in lines 79-84.

Table 1, please explain the data regarding the correlation to viral neutralization for all three CLIAs and add references accordingly.

Answer; you for the insightful comments. All descriptions in Table 1 are provided by the manufacturers. We removed performance data for correlation with viral neutralization from the Table.

Figure 1 Positivity rate at 15-21 days after symptom onset doesn’t seem to reach value >90 for Siemens as stated in the text (page 4 lines 120-121).

Answer; 78 Samples collected 14 days after symptom onset included 38 samples on 15-21 days and 40 samples after 21 days. To clarify this, we added description as “At 15-21 days after symptom onset, Abbott (94.7%), Roche (92.1%) and GenScript (94.7%), except Siemens (86.8%) detected >90% of samples. Overall, for 78 specimens collected 14 days after symptom onset, all four assays detected >90% of samples (Abbott 97.4%, Roche 96.2%, Siemens 92.3%, and GenScript 96.2%).” in lines 124-126.

Pag 4 line 133: “Abbott, Roche, and Siemens assays showed substantial to excellent agreement rates with sVNT…”

Answer; We changed that.

Pag 4 line 140: “Abbott, 5.8-40.000 AU/mL” should be changed with “Abbott, 21.0-40,000 AU/mL”

Answer; We changed that.

Figure 2: add the p value or star (*) in both Abbott and Siemens graphs to indicate the statistical significance.

Answer; Stars (*) are added in Figure 2.

Figure 2: did “severe” samples include “critical” samples?

Answer; Yes, severe cases included severe and critical cases. We clarified that in Figure 2 and added description to result section as “When we compared antibody levels in 19 patients with critical or severe course and 13 patients with mild disease course, only the Abbott and Siemens assay results significantly differed (p < 0.05, Fig. 2, Table S1).” in lines 155-158.

Figure 4:  the y-axis is missing label

Answer; We have changed the Figures.

Page 8 line 213 the concordance between the assays ranged from 0.739 to 0.899, 0.816 represents the agreement rate between Siemens and sVNT.

Answer; Thank you for your careful review, we corrected the error.